# Chemical Profiling, Anticholinesterase, Antioxidant, and Antibacterial Potential of the Essential Oil from *Myrcianthes discolor* (Kunth) McVaugh, an Aromatic Tree from Southern Ecuador

**DOI:** 10.3390/antibiotics12040677

**Published:** 2023-03-30

**Authors:** Diana Romero, Luis Cartuche, Eduardo Valarezo, Nixon Cumbicus, Vladimir Morocho

**Affiliations:** 1Carrera de Bioquímica y Farmacia, Universidad Técnica Particular de Loja (UTPL), Loja 1101608, Ecuador; 2Departamento de Química, Universidad Técnica Particular de Loja (UTPL), Loja 1101608, Ecuador; 3Departamento de Ciencias Biológicas y Agropecuarias, Universidad Técnica Particular de Loja (UTPL), Loja 1101608, Ecuador

**Keywords:** *Myrcianthes discolor*, essential oil, *E*-caryophyllene, anticholinesterase inhibition, MIC, *Enterococcus*

## Abstract

*Myrcianthes discolor,* an aromatic native tree from southern Ecuador, was collected to determine the chemical composition and the biological activity of its essential oil (EO). The EO was obtained by steam-distillation and analyzed by gas chromatography coupled to a mass and a FID detector (GC-MS and GC-FID) and a non-polar DB5-MS column. Enantioselective GC-MS analysis was performed in a chiral capillary column. The antimicrobial, antioxidant, and anticholinesterase potency of the EO was carried out by the broth microdilution method, radical scavenging assays using 2,2′-azino-bis-3-ethylbenzthiazoline-6-sulphonic acid (ABTS) and 1,1-diphenyl-2-picrylhydrazyl (DPPH) radical, and by measuring the inhibition of the acetylcholinesterase (AChE) enzyme. A total of 58 chemical compounds were identified, corresponding to 94.80% of the EO composition. Sesquiterpenes hydrocarbons represented more than 75% of the composition. The main compounds detected were *E*-caryophyllene with 29.40 ± 0.21%, bicyclogermacrene with 7.45 ± 0.16%, *β*-elemene with 6.93 ± 0.499%, *α*-cubebene with 6.06 ± 0.053%, *α*-humulene with 3.96 ± 0.023%, and *δ*-cadinene with 3.02 ± 0.002%. The enantiomeric analysis revealed the occurrence of two pairs of pure enantiomers, (−)-*β*-pinene and (−)-*α*-phellandrene. The EO exerted a strong inhibitory effect against AChE with an IC_50_ value of 6.68 ± 1.07 µg/mL and a moderate antiradical effect with a SC_50_ value of 144.93 ± 0.17 µg/mL for the ABTS radical and a weak or null effect for DPPH (3599.6 ± 0.32 µg/mL). In addition, a strong antibacterial effect against *Enterococcus faecium* was observed with a MIC of 62.5 μg/mL and *Enterococcus faecalis* with a MIC of 125 μg/mL. To the best of our knowledge, this is the first report of the chemical composition and biological profile of the EO of *M. discolor*, and its strong inhibitory effect over AChE and against two Gram-positive pathogenic bacteria, which encourage us to propose further studies to validate its pharmacological potential.

## 1. Introduction

Alzheimer’s disease (AD) is the most common cause of dementia, and it accounts for more than 75% of the reported cases. The pathophysiological process involves synapsis degeneration, mainly characterized by cholinergic neurotransmitter system dysfunction. These characteristics make it possible to face the problem by inhibiting the responsible enzyme of hydrolyzing the neurotransmitter acetylcholine. Galanthamine, huperzine, and other natural derivatives occurring in plants are used for the treatment of AD dementia phase; besides, several plants, including the *Galanthus* genus where the galantamine comes from, produces a bunch of metabolites such as alkaloids, terpenes, polyphenols, and others that have been assessed for their natural AChE inhibitory potential, becoming potential candidates for the discovery of AD alternative treatments [1]. The substrate of AChE, acetylcholine (ACh), is a monoamine (ACh) that plays a fundamental role in central cholinergic neurotransmission, regulating extrapyramidal motor function, cognitive functions, pain perception, and memory [2]. The enzyme acetylcholinesterase (AChE) terminates the effect of ACh at the synapse, regulating its concentration [3]; however, when there is abnormal cholinergic activity, AChE catalyzes in excess the hydrolysis or deactivation of ACh in the cholinergic terminals, causing a decrease/deficit of this neurotransmitter, leading to neurodegenerative diseases [4].

Synthetic AChE inhibitors such as donepezil, rivastigmine, and tacrine are indeed available to treat this type of disease, but they have certain disadvantages, such as gastrointestinal side effects or high costs considering that they are continuous medications [5]. Therefore, there is a great interest in essential oils, which through trials show a potent AChE inhibitory effect [4], in order to develop an effective natural alternative treatment.

Essential oils are a source of secondary metabolites with numerous biological molecules with antibacterial, antifungal, antiseptic, antioxidant, and inhibitory properties of the enzyme acetylcholinesterase (AChE) [1,6,7,8,9,10], of great importance both in the scientific community and in the health area, as possible precursor agents of new phytopharmaceuticals for the treatment of various conditions [11,12]. Species of the genus *Myrcianthes,* belonging to the Myrtaceae family, are wild trees or shrubs, which contain essential oils in their leaves, flowers, or fruits, which have been used within a long tradition to alleviate or treat health problems, and in many other cases, to flavor or season foods and beverages [13].

*Myrcianthes discolor* (Kunth) McVaugh is a native species distributed in the Andes of Ecuador, Peru, Colombia, and Bolivia. It grows favorably in a humid tropical climate, on mountain slopes, between 2900 and 3000 m a.s.l. [14]. There are no extensive reports about its traditional uses, but according to Bussman et al. [15], the fresh whole plant is used to treat inflammation and it was also found to inhibit the growth of *Sthapylococcus aureus.* Its fruit is edible and due to the natural water resistance of its wood is used for building irrigation channels and mill wheels [16]. Despite the lack of information of this species, it is vital for our research group to continue investigating the native flora of Ecuador to find new alternatives for common and emerging diseases. For that reason, our aim was to determine the chemical composition of the essential oil of *M. discolor* and to validate its pharmacological properties through in vitro available assays commonly reported in many publications, such as antioxidant, anticholinesterase, and antimicrobial assays.

## 2. Results

### 2.1. Essential Oil Yield

The essential oil extracted from fresh leaves by steam distillation presented a slightly yellowish viscous appearance and a yield of around 0.075 ± 0.010%.

### 2.2. Qualitative and Quantitative Analyses

The essential oil of *M. discolor* was analyzed by gas chromatography coupled to mass spectrometry (GC-MS) and gas chromatography coupled to flame ionization (GC-FID), using a DB5-MS non-polar column. A total of 58 volatile compounds were identified, corresponding to 94.80% of the total EO of *M. discolor*. Figure 1 shows the GC-MS chromatogram of the EO of *M. discolor* with the retention time of the majority compounds displayed on the X-axis and their relative abundance on the Y-axis.

Sesquiterpenes hydrocarbon accounts for 75.42% of the total composition, with E-caryophyllene (29.40 ± 0.217%), bicyclogermacrene (7.45 ± 0. 162%), *β*-elemene (6.93 ± 0.499%), and *α*-cubebene (6.06 ± 0.053%) as the major compounds. Oxygenated sesquiterpenes were the second representative group with the highest percentage (11.13%) and hydrocarbonated monoterpenes represented 7.42%. Monoterpenes such as limonene (2.63 ± 0.015%), *β*-pinene (1.74 ± 0.010%), and Z-*β*-ocimene (1.69 ± 0.020%) were found in a lower percentage. Table 1 shows the compounds identified according to their elution on the DB5-MS column, and the calculated linear retention index and linear retention as reported in literature [17] are included in this depiction.

### 2.3. Enantioselective Analysis

The enantiomeric composition of the essential oil was analyzed with an enantioselective capillary column 2,3-diethyl-6-tert-butyldimethylsilyl-β-cyclodextrin. Three chiral compounds were identified: (1S,5S)-(−)-β-pinene, (R)-(−)-*α*-phellandrene, found pure with an e.e. of 100%, and the enantiomeric pair (4S)-(−)-limonene and (4R)-(+) limonene in a racemic mixture (e.e. 17.028%) (Table 2).

### 2.4. Antimicrobial Activity

The antimicrobial activity of the essential oil was determined by the broth microdilution method. The highest antimicrobial inhibitory potential of the *M. discolor* EO was displayed by the Gram-positive bacteria *E. faecium* ATCC^®^ 27270, with a MIC value of 62.5 µg/mL, and *E. faecalis* ATCC^®^ 19433 with a MIC of 125 µg/mL. The remaining microorganisms exhibited a weak or absent inhibitory potential at concentrations above 1000 µg/mL. Ampicillin, ciprofloxacin, and amphotericin B were used as positive controls and dimethyl sulfoxide as a negative control. The results of antimicrobial activity are shown in Table 3.

### 2.5. Antioxidant Capacity

The antioxidant capacity of the EO of *M*. *discolor* exhibited weak activity with a value of 144.93 ± 0.1754 and 3599.6 ± 0.324 µg/mL, respectively. Trolox was used as a positive reference control (Table 4).

### 2.6. Anticholinesterase Activity

The EO from the leaves of *M. discolor* exhibited a potent inhibitory effect over acetylcholinesterase (AChE), with an IC_50_ value of 6.68 ± 1.07 µg/mL (Figure 2). Donepezil was used as a positive control with an IC_50_ of 12.40 ± 1.35 µg/mL (Table 5).

## 3. Discussion

The average EO yield was 0.075%, obtained from the leaves of *M. discolor*, which was very low compared to the EOs of *M. fragrans* with 0.28% [18], *M. pungens* with 0.19% [19], *M. mollis* with 0.2%, and *M. myrsinoides* with 0.3% [20] or *M. leucoxyla* with 0.1% [21]. On the one hand, there is relatively little information about the essential oil yield of species from this genus, but according to the literature, the average yield oscillates around 0.1% to 0.3% when obtained from leaves [18,19,20,21,22]. On the other hand, Barra [23] mentioned that endogenous and exogenous factors affect essential oil yields and could even lead to ecotypes or chemotypes in the same plant species, and such factors can involve individual genetic variability, variation among different parts of the plants or phenological stages, and modifications due to the environment. Gupta and Ganjewala [24] suggest that EO can vary due to different factors, such as harvesting conditions, i.e., soil type, cycle in which the plant was found, and part of the plant harvested, or distillation conditions, such as the extraction method or equipment used.

According to the results, it was observed that the EO of *M. discolor* was mostly composed of sesquiterpene hydrocarbons (75.42%) such as *E*-caryophyllene (29.40 ± 0.217%), bicyclogermacrene (7.45 ± 0.162%), *β*-elemene (6.93 ± 0.499%), *α*-cubebene (6.06 ± 0.053%), *α*-humulene (3.96 ± 0.023%), and *δ*-cadinene (3.02 ± 0.002%). Monoterpene hydrocarbons were also found in a lower percentage (7.42%), such as limonene (2.63 ± 0.015%), *β*-pinene (1.74 ± 0.010%), and *Z*-*β*-ocimene (1.69 ± 0.020%). In comparison with studies of related species of the genus *Myrcianthes*, in *M. myrsinoides* (collected in Ecuador), the isomer *Z*-caryophyllene (16.6%) was reported as the main compound, and the monoterpene hydrocarbons limonene (5.3%), *β*-pinene (1.5%), and *α*-pinene (2.5%) in a lower percentage [20]. On the other hand, from the leaves of *M. leuxycola* collected in Colombia, 14 out of 33 compounds were monoterpenes, with 1,8-cineole as the major compound (6.3%) [21]. Other species such as *M. fragrans* were composed mainly by the oxygenated monoterpenes geranial (31.1%) and neral (23.6%) [18], *M. cisplatensis* by hydrocarbonated monoterpenes such as 1,8-cineole (53.8%) and *α*-pinene (16.6%) [25], *M. osteomeloides* and *M. pseudomato* by hydrocarbonated monoterpenes such as 1,8-cineole (55.7% and 24.4%) and *α*-pinene (17.9% and 17.1%), respectively [26], and *M. pungens* by hydrocarbonated sesquiterpenes such as *β*-caryophyllene (11.7%) and 1,8-cineole (10.1%) [19].

*E*-caryophyllene, the main compound of *M. discolor*, is a common sesquiterpene found in essential oils from several species and it has been found to exert some biological activities such as anticancer or analgesic properties [27]. It can cause tracheal smooth-muscle relaxation through voltage-dependent Ca^2+^ channels [28], and Dahham et al. [29] reported a good scavenging radical profile with IC_50_ values of 1.25 and 3.23 µM for DPPH and FRAP assays, respectively, and a more pronounced antibacterial activity against Gram-positive bacteria than Gram-negative, with MIC values ranging from 3 to 9 µM. Additionally, the EO exhibited a moderate AChE inhibitory activity with 32% inhibition at a 0.06 mM concentration [30].

According to the enantioselective analysis, two pairs of pure enantiomers and an almost racemic mixture of limonene (+) and (−) isomers were found. It is worth mentioning that this is the first report of the enantiomeric composition of the EO of *M*. *discolor,* which can be compared with the report of Montalván et al. [20] on the species *M. myrsinoides,* containing the pure chiral compound (+)-limonene with an e.e. of 100% and the enantiomeric pair (+)-*β*-pinene and (−)-*β*-pinene, with an enantiomeric excess of 20.8% (−)-*β*-pinene.

Regarding the biological activity, the EO of *M. discolor* exhibited significant antimicrobial activity against the Gram-positive bacteria of the genus *Enterococcus,* with MIC values less than 125 μg/mL and no activity against the other microorganisms studied at the highest dose tested of 1000 µg/mL. There are no investigations of the antimicrobial activity of the same species, but in comparison with other studies of species of the same genus, Araujo et al. [31] reported that the EO of *M. myrsinoides* presents moderate activity against *B. cereus* with a MIC between 100 and 200 μg/mL, and *B. subtilis* and *S. epidermidis* with a MIC of 200–400 μg/mL. Furthermore, Almeida et al. [19] indicate that the highest antimicrobial activity of *M. pungens* essential oil was against *S. aureus,* with a MIC of 78.12 μg/mL. According to Van Vuuren and Holl [32], the antimicrobial effect of the EO exerted for *E. faecium* and *E. faecalis* can be classified as a strong activity; meanwhile, for the rest of the microorganisms, there is no antimicrobial effect. According to these criteria, the antimicrobial activity of the EO of *M. discolor* can be qualified as good, being a potential source of biological molecules to be studied, which can give rise to new phytopharmaceuticals. We can observe that the species of the same genus possess similar qualities.

Regarding to the antioxidant capacity, the EO of *M. discolor* through the ABTS test displayed a better antiradical capacity with a SC_50_ value of 144.93 ± 0.17. According to Almeida et al. [19], the EO of *M. pungens* leaves has a good antioxidant activity through the DPPH assay, with a value of 24.47 ± 2.03 mg/mL, in contrast to our study where the scavenging capacity through this assay was poor. Selestino et al. [33] mention that EOs rich in sesquiterpenes hydrocarbons, as in the present study, have a low antioxidant activity, which may explain the low activity of the EO of *M. discolor*.

This is the first report of the anticholinesterase activity of the *M. discolor* EO, which exhibited a potent inhibitory effect on the acetylcholinesterase (AChE) enzyme, with an IC_50_ value of 6.68 ± 1.07 µg/mL, which is much higher compared to the value of the *M. myrsinoides* essential oil that has an AChE inhibitory activity of 78.6 μg/mL [21]. This can be explained by the presence of bioactive compounds such as trans-caryophyllene, as suggested by Bonesi et al. [30], and *β*-pinene, *α*-humulene, and limonene, with moderate anticholinesterase activity, as suggested by Hung et al. [34].

## 4. Materials and Methods

### 4.1. Plant Material

The leaves of *M. discolor* in the flowering stage were collected in November and December 2021, on the Loja-Chuquiribamba Road, in the province of Loja, Ecuador, at an altitude of 2560 m a.s. l, with coordinates: 3°56′52″ S, 79°16′11″ W. The plant material was collected under a permit from the Ministry of Environment of Ecuador (MAE-DNB-CM-2016-0048) and identified by Dr. Nixon Cumbicus, and the herbarium sample was deposited in the UTPL Herbarium, with voucher number 14549.

### 4.2. Essential Oil Distillation

The essential oil was obtained from fresh leaves by steam distillation, using a Clevenger-type stainless-steel distiller. The process was carried out for a period of 3 h in each distillation; then, anhydrous sodium sulfate was added to eliminate water residues. It was then stored in sealed amber vials, labeled, and refrigerated at −4 °C for analysis.

### 4.3. Essential Oil Yield

The EO yield, to check how efficient a process is, was calculated by considering the ratio of the volume of EO obtained (*v*/*w*) to the amount in grams of plant matter collected.

### 4.4. Chemical Profiling

Analysis of the EO was performed by gas chromatography coupled to mass spectrometry (GC-MS), and with a flame ionization detector (GC-FID), to qualitatively and quantitatively determine the chemical compounds of *M. discolor*.

#### 4.4.1. GC-MS (Qualitative Analysis)

Chemical analysis of the *Myrcianthes discolor* EO was carried out as suggested by Cartuche et al. [35] with slight modifications, as follows: Samples were prepared by dissolving 990 μL of dichloromethane and 10 μL of EO to reach a 1% (*v*/*v*) concentration. The mass spectrometer was operated in SCAN mode (scan range 40–350 *m*/*z*), with the electron ionization source set to 70 eV, using an Agilent DB-5ms non-polar column, series 122–5532 (5% phenylmethylpolysiloxane, 30 m-long, 0.25 mm internal diameter, and 0.25 μm film thickness). Then, 1 μL of sample was injected, under the following conditions: split mode (split ratio 40:1), using helium as a carrier gas, at a constant flow rate of 1 mL/min. With this column, the thermal program was based on an initial oven temperature at 60 °C for 5 min, followed by a first thermal gradient of 2 °C/min up to 200 °C. Finally, a new gradient of 15 °C/min was applied up to 250 °C, maintaining the final temperature for 5 min. The analysis was performed on a Thermo Scientific Trace 1310 series chromatograph, coupled to a Thermo Scientific ISQ 7000 series mass spectrometer, equipped with a “Chromeleon XPS Software” data system.

#### 4.4.2. GC-FID (Quantitative Analysis)

The analysis technique to quantify the chemical compounds of the EO was performed using an Agilent 6890N series gas chromatograph coupled to a flame ionization detector (FID), employing the same column used for the qualitative analysis (DB5-MS), under the same injection conditions. Quantitative results were recorded as mean values and standard deviation, over three replicates.

The percentage values refer to the weight of the analytes with respect to the mass of the total EO, without using a correction factor.

#### 4.4.3. Enantioselective Analysis

Enantioselective analysis was performed according to the same equipment described above and an enantioselective capillary column (2,3-diethyl-6-tert butyldimethylsilyl-β-cyclodextrin, 25 m-long, 0.25 mm internal diameter, and 0.25 μm film thickness). The thermal program was based on an initial oven temperature of 60 °C for 5 min, followed by a thermal gradient of 2 °C/min up to 220 °C, which was finally maintained for 2 min. The homologous series of n-alkanes (C9–C25) was also injected, which allowed the calculation of the linear retention indices of the stereoisomers.

### 4.5. Antimicrobial Activity

Antimicrobial activity was measured according to the methodology suggested by Cartuche et al. [35]. The values of minimum inhibitory concentration and antimicrobial activity were determined using the broth microdilution technique, with three Gram-positive bacteria (*E. faecalis* ATCC^®^ 19433, *E. faecium* ATCC^®^ 27270, *S. aureus* ATCC^®^ 25923), two Gram-negative bacteria (*E. coli* (O157:H7) ATCC^®^ 43888, *P. aeruginosa* ATCC^®^ 10145), and two fungi (*C. albicans* ATCC^®^ 10231, *A. niger* ATCC^®^ 6275) as microbial test models. The double serial dilution method was used to achieve concentrations ranging from 4000 to 31.25 µg/mL, and a final inoculum concentration of 5 × 10^5^ cfu/mL for bacteria, 2.5 × 10^5^ cfu/mL for yeasts, and 5 × 10^4^ spores/mL were used for sporulated fungi. Mueller Hinton II (MH II) for bacteria and Sabouraud broth for fungi were used as test media.

Commercial antimicrobial agents were used as positive controls as follows: ampicillin solution, 1 mg/mL, for *E. faecalis*, *E. faecium*, and *S. aureus*, ciprofloxacin solution, 1 mg/mL, for *P. aeruginosa* and *E. coli*, and finally, amphotericin B solution, 250 µg/mL, for *C. albicans* and *A. niger*.

### 4.6. Antioxidant Capacity

#### 4.6.1. The 2,2-Diphenyl-1-picrylhydrazyl Radical Scavenging Assay

The DPPH radical scavenging assay was developed according to the methodology proposed by Cartuche et al. [35], using the free radical 2,2-diphenyl-1-picrylhydrazyl (DPPH^-^). A working solution was prepared by dissolving 24 mg of DPPH in 100 mL of methanol and stabilized in an EPOCH 2 microplate reader (BIOTEK, Winooski, VT, USA) at 515 nm until an absorbance of 1.1 ± 0.01 was reached. The antiradical reaction between EO and free radicals was performed at different concentrations of EO (1, 0.5, and 0.25 mg/mL). Then, 270 µL of the DPPH-adjusted working solution and 30 µL of the EO sample were placed in a 96-microwell plate. The reaction was monitored at 515 nm for 60 min at room temperature. Trolox and methanol were used as a positive control and blank control, respectively. The results were expressed as SC_50_ (sweep concentration of the radical at 50%). Measurements were performed in triplicate.

#### 4.6.2. The 2,2-Azinobis-3-ethylbenzothiazoline-6-sulfonic Acid Radical Scavenging Assay

The antioxidant power measured against the ABTS^-+^ cation (2,2′-azinobis-3-ethylbenzothiazoline-6-sulfonic acid) was determined as reported by Cartuche et al. [35], with slight modifications, as described. Briefly, the assay began with the preparation of a stock solution of the radical by reacting equal volumes of ABTS (7.4 µM) and potassium persulfate (2.6 µM) for 12 h, with stirring. The standard solution was prepared by dissolution in methanol to an absorbance of 1.1 ± 0.02, measured at 734 nm on an EPOCH 2 microplate reader (BIOTEK, Winooski, VT, USA). The antiradical reaction was evaluated for 1 h in the dark at room temperature by plating 270 µL of the adjusted ABTS working solution and 30 µL of the *M. discolor* EO at different concentrations (1, 0.5, and 0.25 mg/mL). Trolox and methanol were used as a positive control and blank control, respectively. The results were expressed as SC_50_ (sweep concentration of the radical at 50%). Measurements were performed in triplicate.

### 4.7. Anticholinesterase Assay

AChE inhibition was measured using the spectrophotometric method developed by Andrade et al. [36]. Briefly, the reaction mixture contained 40 μL of Tris Buffer, 20 μL of the analyzed sample solution, 20 μL of acetylthiocholine (ATCh, 15 mM PBS, pH 7.4), and 100 μL of DTNB (3 Mm, Tris Buffer). Preincubation was carried out for 3 min at 25 °C, with continuous shaking. Finally, the addition of 20 μL of 0.5 U/mL AChE initiated the reaction and the amount of product released was monitored on an EPOCH 2 microplate reader (BIOTEK 1) at 405 nm, at 25 °C for 60 min.

The EO of *M. discolor* was prepared by dissolving 10 mg in 1 mL of MeOH. Four further dilutions (10× factor dilution) were included to obtain final concentrations of 1000, 100, and 10 μg/mL. All compounds were assayed at a maximum concentration of 250 μM. Progression curves were calculated from the absorbance according to a standard curve of DTNB and L-GSH at different molar concentrations to measure the rate, expressed as mM/min of product released. As a non-selective protic solvent, MeOH was selected to dissolve the samples and was used as a negative control at a maximum concentration of 10% in the final volume of the mixture, without affecting the enzymatic reaction. Donepezil hydrochloride was used as a positive control with a calculated IC_50_ value of 12.40 ± 1.35 nM.

### 4.8. Statistical Analysis

Data from antioxidant and cholinesterase assays were analyzed through non-linear regression analysis: the log concentration vs. normalized response-variable slope model, by means of the Graph Pad Prism v8.0.1 package (GraphPad, San Diego, CA, USA). SC_50_ and IC_50_ parameters were calculated according to this model. MIC interpretation was in accordance with international standards, such as those suggested by the Clinical and Laboratory Standards Institute, USA, and measured by visual inspection of growth, and due to the method of dilution employed, there were no standard deviations to show. Chemical composition data (%) represented the mean ± SD of three different injections.

## 5. Conclusions

This is the first report of the chemical and biological profile of the essential oil of *Myrcianthes discolor*, which displayed a good antibacterial and strong anticholinesterase effect. Despite its poor antioxidant capacity, similar to other related species from the genus, the information provided in this study encouraged us to propose future scientific research, looking for possible interactions and synergistic or additive effects exerted by the main compounds found in this essential oil. It is uncommon to find an EO that possesses good biological anticholinesterase activity such as that exerted by *M. discolor*, and a possible application could be in a pharmaceutical formulation as an adjuvant for the treatment of AD.

## Figures and Tables

**Figure 1 antibiotics-12-00677-f001:**
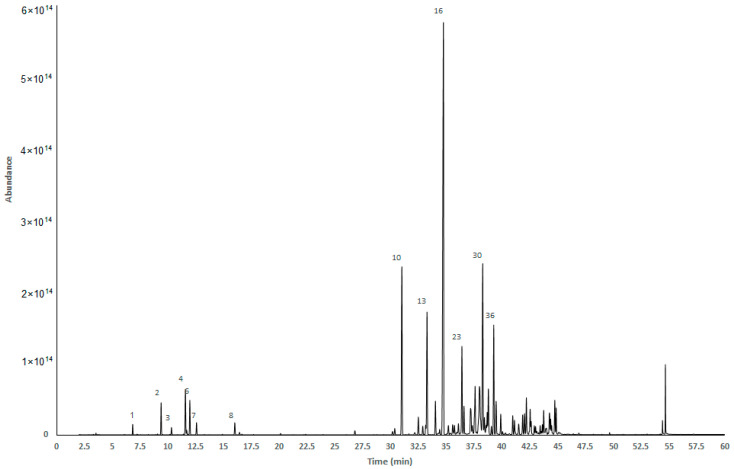
GC-MS chromatogram of *Myrcianthes discolor* EO, through a 5%-phenyl-methylpolysiloxane capillary column.

**Figure 2 antibiotics-12-00677-f002:**
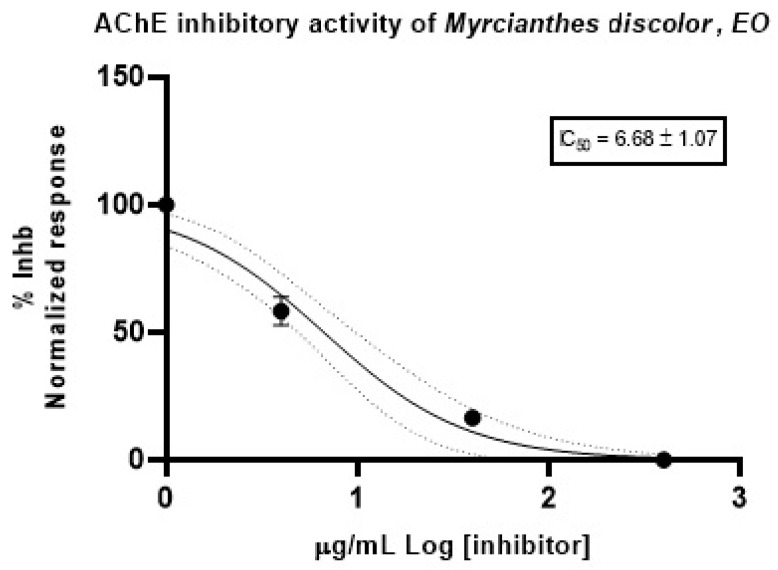
Inhibitory effect plot of the *Myrcianthes discolor* EO, against acetylcholinesterase.

**Table 1 antibiotics-12-00677-t001:** Chemical compounds present in the essential oil of the leaves of *Myrcianthes discolor*.

No.	Compounds	DB5-MS (5%-Phenyl-Methylpolysiloxane)			
		LRI ^a^	LRI ^b^	RA^c^ (%)	CF
1	*α*-Pinene	929	932	0.42 ± 0.002	C_10_H_16_
2	*β*-Pinene	994	974	1.74 ± 0.010	C_10_H_16_
3	*α*-Phellandrene	1012	1002	0.31 ± 0.003	C_10_H_16_
4	Limonene	1033	1024	2.63 ± 0.015	C_10_H_16_
5	*β*-Phellandrene	1035	1025	0.03 ± 0.001	C_10_H_16_
6	(Z)-*β*-Ocimene	1040	1032	1.69 ± 0.020	C_10_H_16_
7	(E)-*β*-Ocimene	1050	1044	0.61 ± 0.004	C_10_H_16_
8	Linalool	1108	1095	0.71 ± 0.010	C_10_H_18_O
9	*δ*-Elemene	1339	1335	0.27 ± 0.004	C_15_H_24_
10	*α*-Cubebene	1351	1348	6.06 ± 0.053	C_15_H_24_
11	*α*-Copaene	1379	1374	0.58 ± 0.002	C_15_H_24_
12	*β*-Cubebene	1391	1387	0.30 ± 0.020	C_15_H_24_
13	*β*-Elemene	1393	1389	6.93 ± 0.499	C_15_H_24_
14	*α*-Gurjunene	1409	1409	1.03 ± 0.006	C_15_H_24_
15	Methyl eugenol	1417	1403	0.21 ± 0.038	C_11_H_14_O_2_
16	*E*-Caryophyllene	1425	1417	29.40 ± 0.217	C_15_H_24_
17	*β*-Copaene	1432	1430	0.04 ± 0.008	C_15_H_24_
18	*β*-Gurjunene	1434	1431	0.29 ± 0.008	C_15_H_24_
19	E-*α*-Bergamotene	1438	1432	0.21 ± 0.045	C_15_H_24_
20	Aromadendrene	1443	1439	0.42 ± 0.003	C_15_H_24_
21	6,9-Guaiadiene	1446	1442	0.24 ± 0.174	C_15_H_24_
22	(Z)-Muurola-3,5-diene	1454	1448	0.39 ± 0.006	C_15_H_24_
23	*α*-Humulene	1461	1452	3.96 ± 0.023	C_15_H_24_
24	9-epi-*E*-Caryophyllene	1465	1464	1.05 ± 0.007	C_15_H_24_
25	Dauca-5,8-diene	1478	1471	1.26 ± 0.033	C_15_H_24_
26	*γ*-Muurolene	1481	1478	0.31 ± 0.002	C_15_H_24_
27	Amorpha-4,7(11)-diene	1487	1479	2.28 ± 0.023	C_15_H_24_
28	*γ*-Himachalene	1483	1481	1.74 ± 0.014	C_15_H_24_
29	Viridiflorene	1490	1496	1.57 ± 0.219	C_15_H_24_
30	Bicyclogermacrene	1502	1500	7.45 ± 0.162	C_15_H_24_
31	*α*-Muurolene	1505	1500	0.43 ± 0.020	C_15_H_24_
32	*δ*-Amorphene	1509	1511	0.54 ± 0.008	C_15_H_24_
33	*E,E*-*α*-Farnesene	1511	1505	1.64 ± 0.419	C_15_H_24_
34	*β*-Bisabolene	1514	1505	1.16 ± 0.038	C_15_H_24_
35	*γ*-Cadinene	1520	1513	0.13 ± 0.009	C_15_H_24_
36	*δ*-Cadinene	1526	1522	3.02 ± 0.002	C_15_H_24_
37	*Z*-Calamenene	1531	1528	0.86 ± 0.009	C_15_H_22_
38	*E*-Cadina-1,4-diene	1541	1533	0.61 ± 0.010	C_15_H_24_
39	Germacrene B	1567	1559	0.64 ± 0.046	C_15_H_24_
40	*E*-Nerolidol	1571	1561	0.15 ± 0.010	C_15_H_26_O
41	Palustrol	1575	1567	0.93 ± 0.021	C_15_H_26_O
42	Spathulenol	1585	1577	0.66 ± 0.015	C_15_H_24_O
43	Caryophyllene oxide	1591	1582	1.09 ± 0.015	C_15_H_24_O
44	Globulol	1595	1590	1.17 ± 0.011	C_15_H_26_O
45	Viridiflorol	1606	1592	0.89 ± 0.008	C_15_H_26_O
46	Guaiol	1608	1600	0.45 ± 0.015	C_15_H_26_O
47	Ledol	1617	1602	0.32 ± 0.010	C_15_H_26_O
48	5-epi-7-epi-*α*-Eudesmol	1620	1607	0.38 ± 0.014	C_15_H_26_O
49	10-epi-*γ*-Eudesmol	1633	1622	0.35 ± 0.035	C_15_H_26_O
50	Junenol	1636	1618	0.12 ± 0.007	C_15_H_26_O
51	*β*-Eudesmol	1639	1649	0.31 ± 0.011	C_15_H_26_O
52	1-epi-Cubenol	1642	1627	0.71 ± 0.013	C_15_H_26_O
53	*Z*-Cadin-4-en-7-ol	1649	1635	0.37 ± 0.012	C_15_H_26_O
54	Cubenol	1658	1645	0.71 ± 0.010	C_15_H_26_O
55	*α*-Cadinol	1661	1652	0.41 ± 0.006	C_15_H_26_O
56	*α*-Muurolol (Torreyol)	1663	1644	0.26 ± 0.010	C_15_H_26_O
57	8-hydroxy-Isobornyl isobutanoate	1672	1674	1.73 ± 0.021	C_14_H_24_O_3_
58	Selin-11-en-4-*α*-ol	1675	1658	0.14 ± 0.040	C_15_H_26_O
	MH			7.42	
	OM			0.71	
	SH			75.31	
	OS			11.13	
	Others			0.21	
	Total			94.80	

^a^ Calculated retention index. ^b^ Retention index on a DB5 column reported in the literature [17]. ^c^ Relative abundance in percentage (%), expressed as mean ± SD (standard deviation). CF: chemical formula; MH: monoterpene hydrocarbons; OM: oxygenated monoterpenes; SH: sesquiterpene hydrocarbons; OS: oxygenated sesquiterpenes.

**Table 2 antibiotics-12-00677-t002:** Enantioselective analysis of the *M. discolor* EO.

No.	Enantiomers	LRI ^a^	ED ^b^ (%)	e.e. ^c^ (%)
1	(1S,5S)-(−)-*β*-pinene	1015	100.00	100.00
2	(R)-(−)-*α*-Phellandrene	1032	100.00	100.00
3	(4S)-(−)-Limonene	1055	58.514	17.028
4	(4R)-(+)-Limonene	1061	41.486	

^a^ Linear retention index. ^b^ Enantiomeric distribution. ^c^ Enantiomeric excess.

**Table 3 antibiotics-12-00677-t003:** Minimum inhibitory concentration calculated for the *Myrcianthes discolor* essential oil against seven human pathogenic microorganisms.

Microorganism	*M. discolor*	Antimicrobial Agent
	MIC µg/mL	
Gram-positive bacteria		Ampicillin (1 mg/mL)
*Enterococcus faecalis* ATCC^®^ 19433	125	0.7812
*Enterococcus faecium* ATCC^®^ 27270	62.5	<0.3906
*Staphylococcus aureus* ATCC^®^ 25923	4000	<0.3906
Gram-negative bacteria		Ciprofloxacin (1 mg/mL)
*Escherichia coli* (O157:H7) ATCC^®^ 43888	-	1.5625
*Pseudomonas aeruginosa* ATCC^®^ 10145	-	<0.3906
Yeasts and sporulated fungi		Amphotericin B (250 µg/mL)
*Candida albicans* ATTC^®^ 10231	4000	<0.098
*Aspergillus niger* ATCC^®^ 6275	-	<0.098

(-) Not active at the maximum dose tested of 4000 µg/mL.

**Table 4 antibiotics-12-00677-t004:** Antioxidant activity of the essential oil of *Myrcianthes discolor*.

EO	ABTS	DPPH
*Myrcianthes discolor*	SC_50_ (µg/mL–µM *) ± SD	
	144.93 ± 0.1754	3599.6 ± 0.324
Trolox *	29.09 ± 1.05	35.54 ± 1.04

* SC_50_: Half scavenging capacity expressed as µM for Trolox.

**Table 5 antibiotics-12-00677-t005:** Anticholinesterase activity of the essential oil of *Myrcianthes discolor*.

EO	Acetilcolinesterasa
*Myrcianthes discolor*	IC_50_ (µg/mL—nM *) *±* SD
	6.68 ± 1.07
Donepezil *	12.40 ± 1.35

* Donepezil IC_50_ is expressed in nanomolar concentration.

## Data Availability

Not applicable.

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
