# Peer review of "Chemical Profiling, Anticholinesterase, Antioxidant, and Antibacterial Potential of the Essential Oil from Myrcianthes discolor (Kunth) McVaugh, an Aromatic Tree from Southern Ecuador"

_antibiotics, 2023, doi:10.3390/antibiotics12040677_

Round 1

Reviewer 1 Report

This manuscript is devoted to an actual topic, but the reviewer has a few comments.

Many researchers, including the authors of this manuscript, have already published similar works. However, this fact should not limit the conduct of such studies.

Overall, although the manuscript contains interesting results, its presentation is very poor. The manuscript looks like a chimera composed of various, unrelated research fragments and contains a significant number of errors. This is likely due to the design of this study.

In the introduction section.

The authors of the referenced literature publication [1] in their manuscript do not consider the use of essential oils in the treatment of progressive neurodegenerative diseases. On the contrary, they are concerned about the AChE inhibitory activity of widely produced and readily available commercial essential oils. Obviously, such an interpretation of these results by the authors of the submitted manuscript is far-fetched and has no basis.

The reviewer considers the authors' phrase “the essential oils of the Myrtaceae family have economic, medicinal and pharmacological properties” (lines 59-60) to be incorrect. I would like to know more precisely what “economic properties” are in relation to essential oils.

In line 63, the term "antileishmanicidal" is most likely a misspelling of "antileishmanial" or "leishmanicidal".

The content of paragraphs in lines 54-58, 59-65, 66-69 is very similar in content, namely, everywhere the possibilities of using Myrtaceae family plants and products from them are redundantly described.

The reviewer believes that the use of the term "vitamin for the brain, memory" (lines 70-72) is incorrect.

Overall, it is not at all clear from the introduction section why the authors chose to carry out this study.

Materials and methods.

The authors used a multicomponent mixture (essential oil) for their research. The reviewer believes that this type of work should use individual substances or use alternative methods for assessing pharmacological activity, such as docking, or something similar.

In subsection 4.5. the antimicrobial activity of the essential oil (lines 288-300) was determined according to [36]; however, there are no such methods in this article. And further. Essential oils are almost insoluble in water. How did the authors obtain such high concentrations of essential oil (hydrophobic) in nutrient broth?

Results section. There is a large table (line 100) that lists the chemical composition of the essential oil. What this table is for is unclear.

It is not clear why compare the antimicrobial activity of an essential oil with the effect of antibiotics. Obviously, antibiotics have a high specificity for specific targets in the microbial cell, and this may be because the antibiotic is an individual substance; essential oil (according to the results of the authors) contains a large number of different substances that can act on various structures of the microbial cell.

The authors report a strong antimicrobial effect of the essential oil against Enterococcus faecalis and E. faecium; but an MIC value of 62.5 or 125 µg/mL is difficult to consider effective.

In the discussion section, the authors discuss the possibility of using essential oils in formulating a potential therapy for improving cognitive function in people suffering from neurodegenerative diseases. The question immediately arises about the possibility of penetration of the components of the studied essential oil through the blood-brain barrier and about the cytotoxicity of these components. The same section discusses the antimicrobial effect of individual substances that make up essential oils, but the authors did not fractionate their product. So you can talk about all the points in this section; the authors compare their results with those of other studies that differ significantly in design.

Authors should be more careful when using cited literature.

The list of references contains many publications not in English (these are numbers 2; 7; 8; 10; 11; 12; 16; 17; 22; 23; 24; 34); it will confuse readers. There are two publications on the list under the same number 36. Authors should reduce the level of self-citation.

Author Response

Dear Reviewer

Reviewer 2 Report

Article paper entitled "A strong antibacterial and anticholinesterase essential oil from Myrcianthes discolor (Kunth) McVaugh from Ecuador ", contain good information about the chemical composition of Myrcianthes discolor essential oil and its pharmacological properties.

     In my opinion, this article can be accepted in its present form.

Author Response

Dear Reviewer 2

Reviewer 3 Report

The current manuscript describes the chemical composition and biological profile of the essential oil of Myrcianthes discolor (Kunth) McVaugh from Ecuador. It appears interesting and worthy of attention. Therefore, the manuscript is suitable to be published after major revisions. Here are my suggestions and comments addressed to the authors:

1. The title should be changed to be adequate with the manuscript content.  It should give enough information about what makes the manuscript interesting.

2. Line 76: “in vitro” should be in italic form.

3. Lines 80-81: this paragraph should have a subtitle as “Essential Oil Yield

4. The subsections should be all in italics and not in bold form, also the subsubsections should not be in bold form. The authors sometimes capitalized the first letter of words and sometimes did not, all should be the same, please check all sections, subsections and subsubsections and make them as required.

5. In the “Antimicrobial activity” subsection, the MIC values were determined! what about Minimum Bactericidal/Fungicidal Concentration (MBC/MFC)???

6. Tables 4 and 5 should be as in the template.

7. The authors should describe the properties of the major compounds identified in essential oil and explain their effects on the biological activities evaluated.

8. Materials and Methods: The subsection on the description of data analysis/statistical analysis is missing, please add this part.

9. Lines 181-193: the reference [28] is missing in-text citations!

10. They are some errors and mistakes in text writing, please check all and correct them. 

Author Response

Dear Reviewer 3

Reviewer 4 Report

Search for safe compounds is very urgent to overcome the proplem related to microbial infection and other diseases

The following comments don't lose any benefits from the work 

  1- Giving the full name of any abbreviation in the abstract such as DPPH, etc

2- why not  determine MBC to determine the cidal or static effect of the extract 

3-Antimicrobial activity, where the inhibition zone  at each microbe till to determine activity against Aspergillus 

4-  please classify the activity of extract : good, moderate, and weak if the MIC  is less than  100 µg/mL, from 100 µg/mL to 625  µg/mL, and more than 625  µg/mL, respectively. 

5- Discussion needs to rewrite its very long compared with the result

6- cytotoxicity experiment is very important : if possible  where  this experiment 

7- Is this This  the first report of the chemical composition and biological profile of the essential oil of Myrcianthes discolor as mentioned in conclusion

8- avoid the similarity between abstract and the conclusion 

Author Response

Dear Reviewer 4

Round 2

Reviewer 1 Report

The authors have done a tremendous amount of work to improve this manuscript and, apparently, it can be accepted in its present form. The only comment of the reviewer is not of a fundamental nature, it concerns the description of statistical procedures in somewhat unexpected places, see, for example, lines 349-351: "The corresponding IC50 value was calculated by fitting the data to the curve (linear regression or nonlinear regression analysis, PRISM 8.0.1, GraphPad, San Diego, CA, GraphPad, San Diego, CA, USA) Wouldn't it be better to move such text snippets to the statistical methods subsection?

Author Response

Dear editor

The answers to reviewers are attached in a file.

Regards!

Reviewer 3 Report

The manuscript is much better after major revision, I appreciated your efforts to improve the manuscript. However, still only a few corrections:

1. Line 21: Please correct “broth microdilution method” instead of “microdilution broth method”

2. Line 81: “M. discolor” should be in italics.

3. Line 82: “in vitro” should be in italics.

4. Line 122, 133:  Please correct “Table” instead of “table”

5. Line 140: Please add the units for “144.93 ± 0.1754 and 3599.6 ± 0.324”

6. The word “Table” in text, sometimes is written in bold sometimes not. Please make them appropriate.

7. Line 147: “Figure” instead of “figure”

8. Line 208: “genus” should be not in italic.

Author Response

(The authors gave the same response as above.)

Reviewer 4 Report

 Accept in present form

Author Response

(The authors gave the same response as above.)
